# Application of Ethnobotanical Indices in the Utilization of Five Medicinal Herbaceous Plant Species in Benin, West Africa

**Hubert Olivier Dossou-Yovo** [1,*] **, Fifanou Gbèlidji Vodouhè** [2] **, Alevcan Kaplan** [3] **and Brice Sinsin** [1]

1   Laboratory of Applied Ecology, Faculty of Agronomic Sciences, University of Abomey-Calavi Benin, Godomey P.O. Box 1974, Benin; bsinsin@gmail.com
2   Laboratory of Economic and Social Dynamics Analysis (LARDES), Faculty of Agronomy, University of Parakou, Parakou P.O. Box 123, Benin; fifanou.vodouhe@fa-up.bj
3   Department of Crop and Animal Production, Sason Vocational School, Batman University, Batman 72060, Turkey; kaplan.alevcan@batman.edu.tr
*   Correspondence: dohuoly@yahoo.fr; Tel.: +229-97957040

**Abstract:** The ethnobotanical utilization of five neglected herbaceous species, *Argemone mexicana* L., *Heliotropium indicum* L., *Kedrostis foetidissima* (Jacq.) Cogn., *Peperomia pellucida* (L.) Kunth and *Schrankia leptocarpa* DC. was investigated in Southern Benin to determine the ethnomedicinal and magic knowledge on them. Thirty-six herbal medicine traders were surveyed in six different markets in three districts. Four ethnobotanical indices were used. All informants traded *A. mexicana* and the majority traded *H. indicum*, *K. foetidissima*, and *P. pellucida*. Purchases in the traders' own markets was the single most important source of *H. indicum*, *A. mexicana* and *P. pellucida*. *A. mexicana* was the most demanded by customers. Traders reported the scarcity of *A. mexicana* and *H. indicum* and the availability of *S. leptocarpa*, *K. foetidissima* and *P. pellucida*. *H. indicum* was mainly used to treat hypertension and fever. Similarly, *S. leptocarpa* was mostly mentioned in the treatment of hypertension and to facilitate childbirth. *K. foetidissima* mainly served religious and animist purposes. Similarly, *P. pellucida* was reported as being mainly used to implant a vodun, a traditional religion in West Africa. *A. mexicana* served to treat babies just after the umbilical cord fall as well as jaundice. *S. leptocarpa* and *P. pellucida* exhibited the highest Use Value (UV), and there was a very low similarity between study species in terms of uses. The majority of traders did not plant the study species, although they serve to treat various social conditions. We suggest a better management of *H. indicum* and *S. leptocarpa* through collection for trading and medicinal utilization while the planting is required for *A. mexicana* and *P. pellucida* because of their scarcity. *K. foetidissima* should be preserved and used as medicine wherever it occurs.

**Keywords:** Benin; ethnobotanical indices; ethnopharmacology; neglected herbaceous

## 1. Introduction

Medicinal plants are valuable sources of herbal products [1], and herbal medicine is of major importance to mankind [1–4]. Medicinal plants also serve as raw materials for pharmaceutical factories [5]. As a result, there is an increasing need to gather knowledge related to ethnomedicine in all parts of the world. The exploitation of medicinal plants frequently leads to threats towards their natural populations. For instance, Leaman [6] reported that some plants known as medicines years ago have become extinct. Moreover, Catarino et al. [7] stated the threats that the overexploitation of native legume species for medicinal purposes constitute. Bekalo et al. [8] suggested the creation of measures to conserve medicinal plants in natural ecosystems and other suitable environments. Indigenous people's interest in the use of plants has always been based on the assumption of species availability on a continuing basis. The medicinal exploitation as well as conservation of medicinal plants should be a priority during scientific investigations. Many plants are used as medicines but neglected by research all over the world.

In Africa, traditional medicine practitioners (TMPs) or traditional healers are diagnosticians as well as those who prescribe herbal medicines. Rural communities with access to modern medical services have a strong belief and respect for traditional healers who provide first aid for most people in remote areas [9]. Traditional medicinal plants are widely used in Benin as well as in different parts of Africa. In the Republic of Benin, researchers investigated many medicinal species but the gap persists on many other plants [10,11]. Furthermore, Dossou-Yovo et al. [12], stating the medico-magic knowledge on four herbaceous species highly used in ethnomedicine throughout Benin, highlighted that much research attention should be focused on herbaceous medicinal plants in Benin. However, to the best our knowledge, in general, *A. mexicana*, one of the five medicinal plants constituting the herbal material of our research, is a plant whose homeland is tropical America, and which spreads in tropical and subtropical regions of the world. In the conventional system of medicine, the entire plant of *A. mexicana* is extensively used in the treatment of tumors, warts, skin diseases, inflammations, rheumatism, jaundice, infections, leprosy, piles, warm infestations and dysentery; it also has anti-HIV activity in H9 lymphocyte (with $EC_{50}$ value: 1.77 µg/mL; Therapeutic Index: 14.6). The boiled leaves of the plant are used to treat malaria fever and ulcers. The seeds are effective in the treatment of leprosy, dropsy and jaundice. The juice of the plant is used in the treatment of scorpion bites [13–15]. Most of the compounds that have a therapeutic effect in the treatment of diseases belong to the class of alkaloids, although there are also terpenoids, flavonoids, phenolics, long chain aliphatic compounds and a few aromatic compounds [14]. All parts of the *H. indicum* plant are claimed to have medicinal properties. The entire plant contains alkaloids such as heliotrin, laciocarpin, indisine, acetyl indices, indisine, indisine-N-oxide, retronesin, and trachelantamide [16]. The leaves are used to treat ophthalmic disorders, erysipelas and pharyngodynia and its roots are used as an astringent, expectorant and antipyretic [16]. The aqueous extract of the leaves has been proven to be active against Schwart's leukemia [17]. Moreover, in some studies, it has been determined that the plant has antituberculosis (MIC value as 20.8 µg/mL), antibacterial, and gastric protective effects [18–20]. Many scientific reports show that crude extracts and extensive numbers of phytochemical constituents (therapeutic phytoconstituents were phenols, alkaloids, flavonoids, tannins, terpenoids and steroids) isolated from *K. foetidissima* have activities including antimicrobial, antioxidant, anticancer, gastroprotective, anti-inflammatory and various other important medicinal properties [21]. In Elavazhagan and Balakrishnan's [22] study, the effects of some extracts found in the leaves, stems and fruits of *K. foetidissima* against *Pseudomonas* sp., *Staphylococcus* sp., *Bacillus* sp., *Vibrio cholera*, *E. coli*, *Lactobacillus brevis*, *Lactobacillus bulgaricus*, *Micrococcus luteus* and *Proteus discteus* were investigated by the diffusion method. They found that this technique, a semi-numerical method, is advantageous in terms of serial dilution and easy interpretation was effective against the bacteria tested, and the maximum and minimum inhibition zones were *P. aeruginosa* (10 mm) and *V. cholera* (3 mm) on the leaf, respectively. *P. pellucida* extracts, fractions or isolated components have been shown to have analgesic, anti-inflammatory, antipyretic, antioxidant, antihyperglycemia, antihyperuricemia, burn healing, anti-fever and headache, antidepressant effect, gastric protective, hypotensive, cytotoxic activities, antimicrobial, human cervical cancer cell line (HeLa), human liver cancer cell line (Hep G2), generally cell disease preventive, lipase inhibitor. However, it has fibrinolytic and thrombolytic, antidiarrheal, antiosteoporotic, and antihyperglycemia activities [23–26]. On the other hand, *S. leptocarpa* is a perennial herb native to tropical South America and was introduced to West Africa, where the species is traditionally used in folk medicine [27]. *S. leptocarpa* is used to treat eruptive fever, hypertension (aerial parts of the plant boiled), jaundice, infections, abdominal pains, hiccup, malaria and has both antioxidant effect ($IC_{50}$ value of DPPH activity: 1.35 µg/mL) and reduction in blood pressure of Wistar rats [28–31]. All of the chemical potentials stressed on the study species show their importance in pharmacology and therefore the relevance of this ethnobotanical and conservation research.

It is clear that the medicinal plants in question have many ethnomedicinal, animist and pharmaceutical uses in many regions, and in the southern part of Benin these plants are frequently used for medicinal purposes in daily life. However, it is important to highlight that none of the five study species is native to Benin, where they are alien plants, with *S. leptocarpa* being invasive and the remaining four being weeds [32]. *H. indicum* is an annual weed, *A. mexicana* is becoming rare throughout Southern Benin, *P. pellucida* is a weed somehow rare due to its specific ecological areas comprising the surroundings of habitations, humid areas and commercial gardens, while *K. foetidissima* is a spontaneous rare weed in Southern Benin (Personal observation and communication with Prof. Aristide C. Adomou, botanist at The National Herbarium of the University of Abomey-Calavi). In fact, plant species can serve as medicines no matter what origins they have. For instance, plants not native to Benin but found in relation with termitaria were exploited as medicines in the Pendjari Biosphere Reserve in the northern part of the country [3]. In addition, mangrove dwellers exploited plants not native to Benin for medicinal purposes [4]. Likewise, plan species not locally native, were recorded in the treatment of candidiasis in the southern region of the country [11]. Despite their importance in folk medicine, the study species are less researched locally. For this reason, we conducted this study, which offers a new perspective in terms of recognition, better management, and the protection of these herbaceous plants, which are always sold in herbal medicine markets in Benin. To do this, in order to assess the diversity of medico-magic knowledge related to these species, four ethnobotanical indices recently used by Dossou-Yovo et al. [12], Relative Frequency Citation (RFC), Fidelity Level (FL), UV and Rahman Similarity Index (RSI), were applied. Ethnobotanical indices are recognized as relevant for quantitative and qualitative ethnobotany [33,34]. In this respect, we believe that this study will be the basis for many future studies.

## 2. Materials and Methods

### 2.1. Study Area

Surveys were undertaken in some of the most populated towns of southern Benin (West Africa) with herbal medicine traders in these markets. These were the Pahou, Zobê and Kpassê markets in the Ouidah District, with 445 inhabitants/sq km (the Atlantic Department), the Cococodji and Godomey markets in the Abomey-Calavi District, 1010 inhabitants/sq km (the Atlantic Department), and the Vêdoko and Dantokpa markets in the Cotonou district with 8595 inhabitants/sq km (Department of Littoral). Figure 1 shows the location of the surveyed cities in Southern Benin where the predominant local language is Fon.

### 2.2. Vegetation, Soil Types and Climate of the Surveyed Areas

Adomou [32] distinguished three major phytochorological zones in Benin, namely the Guineo-Congolian and Sudanian regions, linked by the Guineo-Sudanian transition zone. The areas surveyed in this study (Ouidah, Abomey-Calavi and Cotonou) and located in Southern Benin and belong to the Guineo-Congolian phytochorological region. The Guineo-Congolian endemic genera such as Amphimas, Anthonotha, Distemonanthus, Hymenostegia, Anthrocaryon, Coelocaryon, and Discoglypremna were reported by Adomou [32]. The only Guineo-Congolian endemic family found in Benin is Octoknemataceae with *Octoknema borealis* (now in Olacaceae), restricted to the Phytodistrict of Ouémé valley. The natural vegetation of Southern Benin is dense dry semi-deciduous forest but is currently degraded [35]. With regard to the soil types of the surveyed areas, it is important to highlight that Southern Benin up to the latitude of approximately 6°50′ is dominated by the Terre de Barre, which is a ferralitic soil developed in the continental terminal [35]. The "continental terminal" constitutes part of the continental deposits which in Africa have a very large geographic extension and a high thickness. These deposits belong to all geological periods. They were formed from the Precambrian era and are still found in the Pleistocene. Similarly, Dossou-Yovo et al. [4] stated that the most significant soil types

in the surveyed areas are sandy soils, hydromorphic soils, and ferralitic soils. Figure 1 highlights more details related to the vegetation and soil distribution in the study area. The climate of the surveyed cities is characterized by two rainy seasons, from April to July and October to November, and two dry seasons, from August to September and December to March [4]. They reported an annual precipitation ranging from 820 to 1300 mm, and the annual average temperature is about 33 °C.

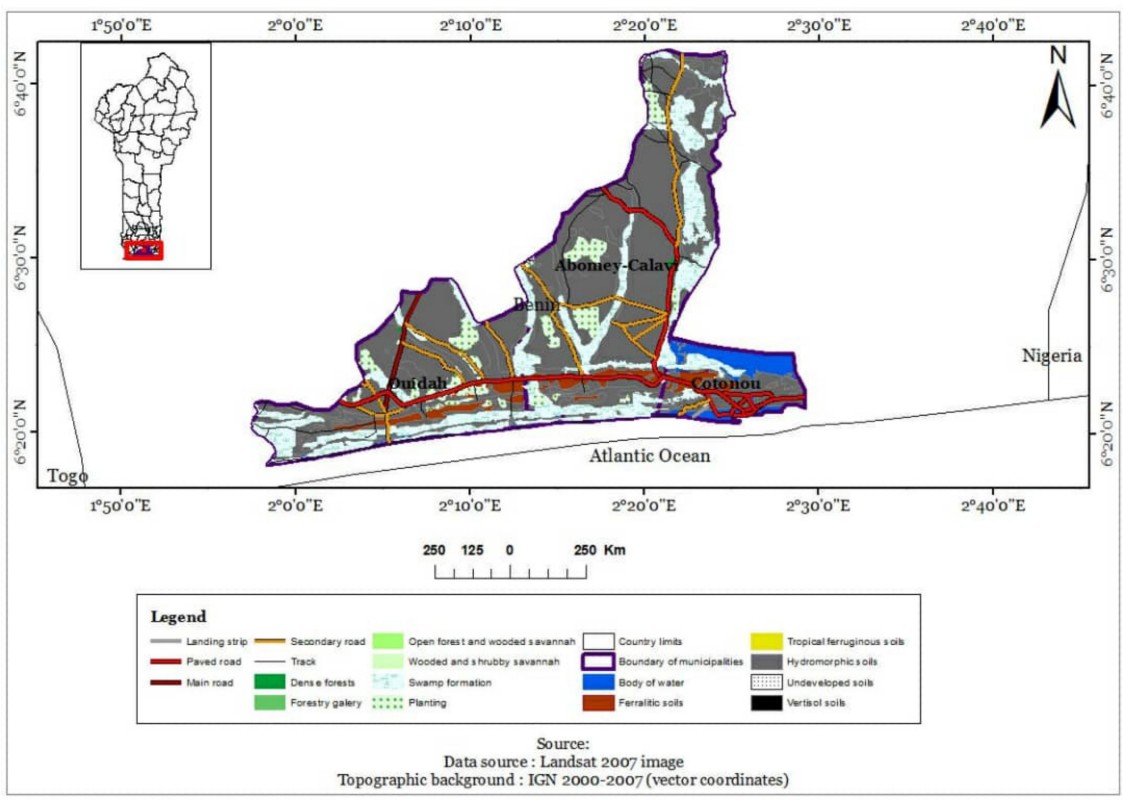

**Figure 1.** Map showing the location of the surveyed districts in Southern Benin. Location of the surveyed districts in Southern Benin, vegetation and soil distribution.

### 2.3. Plant Materials

The plant materials used in the study are given in Figure 2. *A. mexicana*, belonging to the Papaveraceae botanical family, is an erect annual herb that grows up to 1 m with a long, slightly branched taproot. The stem is cylindrical to oblong, smooth, and pale greenish in color (www.prota.org accessed on 19 June 2022). The entire stem is covered with very short hairs and few long yellowish spines. It is named Houêtchégnon in the local language of Fon and worldwide is known as the Mexican prickly poppy, prickly poppy or yellow prickly poppy in English. The species is well known and is used as medicine in Benin as reported by de Souza [36]. A voucher specimen was also certified and deposited at the National Herbarium of Benin at University of Abomey-Calavi (Voucher Number: YH 519/HNB). *H. indicum* (Boraginaceae) is an annual, erect, branched plant that can grow to a height of about 15–50 cm (5.9–19.7 in). It has a hairy stem, bearing alternating ovate to oblong-ovate leaves. It has small white or purple flowers with a green calyx; five stamens borne on a corolla tube; a terminal style; and a four-lobed ovary (www.prota.org accessed on 19 June 2022). It is locally called Koklosou dinkpadja in thel Fon language, and worldwide it is known as Indian heliotrope, or Indian turnsole in English. The species is well known and medicinally used in Benin, as reported by de Souza [36]. A voucher specimen was certified and deposited at the National Herbarium of Benin at the University of Abomey-Calavi (Voucher Number: YH 520/HNB). *K. foetidissima* (Cucurbitaceae) is an herb with prostrate or climbing stems up to 3 m long, growing from a tuberous root. The

plant is somewhat sticky and unpleasantly smelling when crushed. Leaves are narrowly to broadly ovate in outline (www.zimbabweflora.co.zw accessed on 19 June 2022). It is named Tchiôman or Donkpêman in Fon, a medicinal plant used in Benin for decades, as has been reported by de Souza [36]. A voucher specimen was also certified and deposited at the national herbarium of Benin at University of Abomey-Calavi (Voucher Number: YH 521/HNB). *P. pellucida* (Piperaceae) is an annual herb up to 0.4 m tall, its leaves are broadly ovate (egg-shaped) or ovate-triangular. It is branched, and its hairless stems are erect or rising upwards (https://www.nparks.gov.sg/florafaunaweb/flora/5/5/5557 accessed on 19 June 2022). The species is named Sinfama in Fon and is known worldwide as pepper elder, shining bush plant or man to man. It was also reported by de Souza [36]. A voucher specimen was certified and deposited at the National Herbarium of Benin at University of Abomey-Calavi (Voucher Number: YH 518/HNB). *S. leptocarpa* (Leguminosae-Mimosoideae) is a straggling perennial herb resembling the "Sensitive Plant" (*Mimosa pudica*) but with continuous not jointed pods (www.plants.jstor.org accessed on 19 June 2022). It is locally named Assoutowédja boyonoudo, is a medicinal plant used throughout Benin, and was reported by de Souza [36]. A voucher specimen was certified and deposited at the National Herbarium of Benin at University of Abomey-Calavi (Voucher Number: YH 522/HNB). All of these plant names with their photographs were checked with http://www.theplantlist.org (accessed on 19 June 2022).

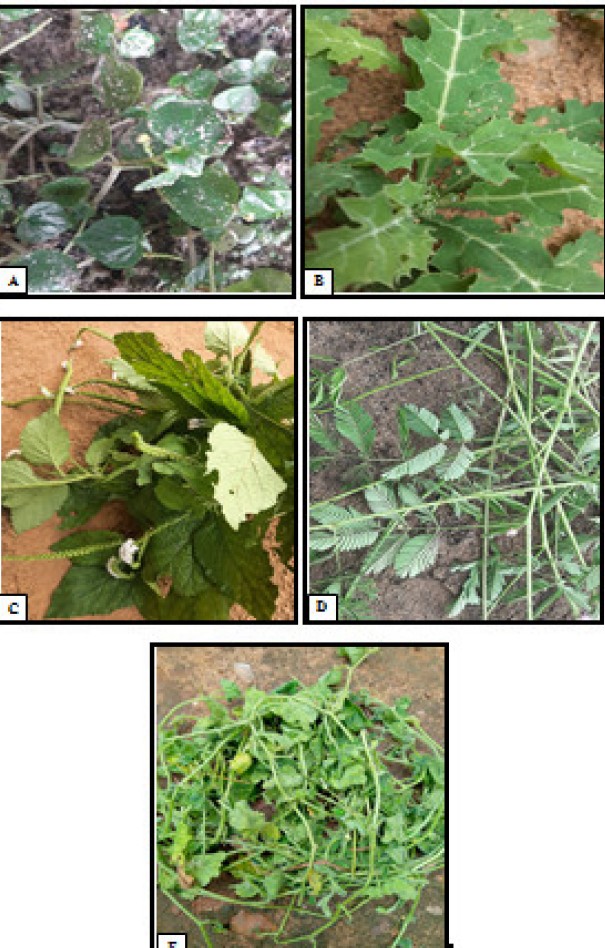

**Figure 2.** Photograph showing plant materials used in the study, (**A**) *P. pellucida* (**B**) *A. mexicana* (**C**) *H. indicum* (**D**) *S. leptocarpa* (**E**) *K. foetidissima*.

With regards to the origin and ecological conditions of the research species, it is important to emphasize that all of them are not native to Africa. In fact, *A. mexicana* was

originally found in Mexica (South America) and it is now naturalized in many parts of the world including Africa. *A. mexicana* is adapted to a wide range of habitats. It is reported as occurring mainly in regions with a distinct dry season. It occurs as a weed of arable land, pastures and in waste places, roadsides, railways and fence rows.

*H. indicum* is native to Asia, a common weed in wastes and settled areas in many parts of the world including Africa. *K. foetidissima* has a native range from Tropical and Southern Africa, and South Western Arabian Peninsula. The species grows in the wild in rain-forest and river margins; deciduous and semi-evergreen woodland and dry bushland, wooded grassland; termite mounds; and at elevations from sea level to 2000 m. *P. pellucida* has a native range of tropical and subtropical America, tropical Africa, and Madagascar. It grows well in full to moderate light. This plant also grows under fluorescent lights, which makes it ideal for office or commercial buildings. This information is available on the website of the Royal Kew Botanic Garden (https://powo.science.kew.org/taxon/urn:Isid:ipni.org accessed on 19 June 2022). *S. leptocarpa* is a species native to the Tropical South America and introduced in West Africa (Ghana, Benin and Nigeria) (www.plants.jstor.org accessed on 19 June 2022).

### 2.4. Ethnobotanical Data Collection

The proposal to conduct ethnobotanical investigations on herbaceous species through market surveys was submitted to and approved by the Scientific Council of the University of Abomey-Calavi, and by the Laboratory of Applied Ecology, Faculty of Agronomic Sciences, University of Abomey-Calavi, Benin, using the framework of our previous research [12]. The aim of the research was explained to the official responsible of each market from whom the approval to conduct the surveys was sought. Then the verbal consent was obtained from all the market responsibles. Similarly, the aim of the study was explained to each trader from whom we obtained the verbal consent to participate in the research.

Two markets in each of three districts of Benin were surveyed, Pahou and Zobê in the Ouidah District, Cococodji and Godomey in the Abomey-Calavi District, and Vêdoko and Dantokpa in the Littoral District. Six herbal medicine traders were randomly selected in each market and ethnobotanical data were gathered from them on each of the five study species. A total of 36 traders were thus surveyed. A semi-structured questionnaire was used to collect data on the trade, sources and importance of each species. Traders were asked to rank the five species according to consumer demand regardless of their uses [12,37], and the availability versus scarcity of each species in recent years. The various causes of scarcity were also recorded. In addition, herbal medicine traders were asked to list all diseases, disorders and magic issues treated using each species as well as all types of preparations used for these purposes. To assess traders' contribution to species conservation, the plantation or in situ conservation of each species by traders and their motives were investigated. Each time an informant confirmed the trade of a species, it was asked to show a sample for positive identification. All interviews were conducted in the local language of Fon, which was spoken and understood by all informants.

### 2.5. Ethnobotanical Indices Calculation for Data Analysis

(a).   The Relative Frequency Citation (RFC) [38,39] was calculated using the formula:

$$\text{RFC } (\%) = \frac{n_i}{N} \times 100 \tag{1}$$

where $n_i$ is the number of informants who mentioned the concerned species, thing or aspect and N is the total number of traders surveyed.

(b).   The Fidelity Level (FL) [40–42] was determined for the diseases as follows:

$$\text{FL } (\%) = \frac{I_P}{I_u} \times 100 \tag{2}$$

where $I_p$ is the number of informants who mentioned the use of a species for a specific disease, disorder or magic issue and $I_u$ is the total number of informants who mentioned the species for any use. FL was also calculated in a similar way for the types of preparation recorded for each disease, disorder or magic issue. The FL served to assess the informants' preference for a species to treat a specific disease, disorder or magic issue. It also served to assess the preference for a specific type of preparation. FL ranges from 0 to 100% and a value close to 100% indicates a strong preference.

(c). The UV of each species was determined [38,39,43] by the equation:

$$UV = \frac{n_d}{I_u} \tag{3}$$

where $n_d$ is the total number of use reports (diseases, disorders or magic utilization) mentioned for a species and $I_u$ is the total number of informants who mentioned the species for any use. The UV served to compare the relative importance of the study species in terms of uses.

(d). The ethnobiological Rahman Similarity Index (RSI) [44], to assess the similarity between species in terms of uses, was calculated as follows:

$$RSI\ (\%) = \frac{n_c}{n_a + n_b - n_c} \times 100 \tag{4}$$

This formula is similar to the Jaccard Similarity Index. Rahman et al. [44] considered a social condition recorded in two communities of people and treated with a certain number of medicinal plant species by each community as well as the number of plant species commonly used for this treatment in both communities. In the present research, this approach was adapted to similarity of uses of pairs of species with, respectively, $n_a$ and $n_b$ number of use reports in species a and b, respectively, and the number of common uses recorded for the two species as $n_c$. RSI can range from 0 to 100% and a RSI lower than 50% means low similarity in terms of uses between the two species while a RSI higher than 50% indicates a high similarity of uses between the two species.

## 3. Results and Discussion

All respondents confirmed the trade of *A. mexicana* while the majority of them, 94%, 88%, and 72%, respectively, traded *H. indicum*, *K. foetidissima* and *P. pellucida*. *S. leptocarpa* was traded by 55% of the surveyed traders. Seven sources of the species traded were recorded and the majority of the traders purchase *H. indicum* and *A. mexicana* from their own markets, while 50% of informants also purchase *S. leptocarpa* and *P. pellucida* from their own markets (Table 1). A relatively large proportion of respondents purchase *K. foetidissima* from other distant markets. Furthermore, noticeable proportions of herbal medicine traders also purchase *H. indicum* and *A. mexicana* from other distant markets, and collect *S. leptocarpa* from the wild populations for trade. The purchase from their own markets was the single most important source reported by traders for *H. indicum* (44%), *A. mexicana* (50% of traders), and *P. pellucida* (33% of traders). Both purchases in their own markets and collection from the wild were predominant for *S. leptocarpa* (22% each). A low proportion of traders (25%) reported the purchase from other distant markets as the most important source for *K. foetidissima*.

Regarding consumer demand, *A. mexicana* was ranked as first by 44% of traders followed by *K. foetidissima*, which was mentioned by 33% of traders (Table 2). The third most demanded species was *H. indicum* as reported by 22% of traders. *P. pellucida* was the fourth at 33%. The least traded species was *S. leptocarpa*. The majority of traders reported the scarcity of *A. mexicana* in recent years. Similarly, half of surveyed traders mentioned the scarcity of *H. indicum*. Contrary to these perceptions, noticeable proportions of respondents reported the availability of *S. leptocarpa* (44% of traders), *K. foetidissima* (50% of traders), and *P. pellucida* (39% of traders). Four factors causing scarcity were reported by herbal medicine traders. Climate change as well as the destruction of the species habitats for

logging was mainly cited for *H. indicum* and *A. mexicana* (Table 3). Five factors contributing to availability of the species were mentioned. The ruderal character was predominantly reported for *S. leptocarpa* while the abundance of rains, wholesalers of medicinal plants and planters were equally recorded for *K. foetidissima*. Similar to this, the predominant factor of availability reported by traders for *P. pellucida* was the widespread cultivation in commercial gardens.

**Table 1.** Sources of the research species according to traders.

| Plant Samples | Purchase in Their Markets | Purchase in Nearby Markets | Purchase in Distant Markets | Purchase in Gardens | Collection from Wild Populations | Collection from Own Gardens | Collection from Natural Populations at Home |
|---|---|---|---|---|---|---|---|
| | RFC (%) | | | | | | |
| *H. indicum* | 61 | 33 | 44 | 5 | 38 | 5 | 11 |
| *S. leptocarpa* | 50 | 11 | 16 | 0 | 33 | 0 | 0 |
| *A. mexicana* | 72 | 33 | 50 | 5 | 0 | 5 | 5 |
| *K. foetidissima* | 33 | 11 | 55 | 0 | 11 | 5 | 0 |
| *P. pellucida* | 50 | 0 | 5 | 11 | 0 | 16 | 5 |

**Table 2.** Ranking of the study species according to their traders.

| Plant Samples | RFC (%) | | | | |
|---|---|---|---|---|---|
| | 1st Rank | 2nd Rank | 3rd Rank | 4th Rank | 5th Rank |
| *H. indicum* | 22 | 22 | 16 | 22 | 11 |
| *S. leptocarpa* | 0 | 5 | 5 | 11 | 33 |
| *A. mexicana* | 44 | 27 | 27 | 0 | 0 |
| *K. foetidissima* | 33 | 22 | 33 | 0 | 0 |
| *P. pellucida* | 0 | 22 | 5 | 33 | 11 |

**Table 3.** Factors causing the scarcity of plant samples reported as quite rare by traders.

| Plant Samples | Habitat Destruction for Logging | Climate Change | Decrease in Soil Fertility | Overexploitation of the Species |
|---|---|---|---|---|
| | RFC (%) | | | |
| *H. indicum* | 16 | 22 | 5 | 5 |
| *A. mexicana* | 11 | 44 | nd | nd |
| *S. leptocarpa* | nd | nd | nd | nd |
| *K. foetidissima* | nd | nd | nd | nd |
| *P. pellucida* | nd | nd | nd | nd |

nd (not determined): Indicates that these species are not concerned.

The majority of informants confirmed the trade of the research species. This proves the importance of those species in herbal medicine in Benin. Regarding the purchase of *H. indicum*, *A. mexicana* and *P. pellucida* in the traders' own markets, the authors conclude that medicinal plants collectors and wholesalers can more easily obtain those species than the other ones. The most common collection of *S. leptocarpa* from the wild populations by herbal medicine traders indicates a connection with their natural environment. The purchase of *K. foetidissima* in more distant markets shows that the species is not easily found and collected locally by harvesters, wholesalers or informants.

The ranking of *A. mexicana*, *K. foetidissima* and *H. indicum* as the most demanded reveals that these species play a key role in ethnomedicine within the country. As a result, their exploitation constitutes a high pressure on natural populations, which was confirmed by the scarcity of *A. mexicana* and *H. indicum* reported by the majority of informants. Although mostly purchased in more distant markets, and most demanded, *K. foetidissima*

was reported as relatively available in recent years, as were *S. leptocarpa* and *P. pellucida*. There is an urgent need to define conservation tools and strategies for *A. mexicana* and *H. indicum* by promoting their planting in homes and commercial gardens. But taking into account the ecology of *H. indicum* as a weed, we assume in this study that the species should be globally available in rural areas where fields and fallows are abundant. Similarly, *P. pellucida* must also be promoted through conservation programs as it is a rare species in Southern Benin and informants reported its planting in commercial gardens. *S. leptocarpa* is invasive, meaning very common in fields, fallows and other areas throughout the country. Therefore, herbal medicine traders should be trained and sensitized to use the populations of this species for personal medico-magical and trading purposes wherever it occurs.

Remarkably, examining the diversity of uses, parts, preparations and routes of administration of plants revealed valuable information. Twelve ethnobotanical uses of plant samples (two for spiritual and ten for treating diseases) were recorded (Table 4). For *H. indicum*, the uses cited by the greatest number of herbal medicine traders were the treatment of hypertension (FL; 47%), and fever (FL; 29%). The most commonly used parts in the treatment of hypertension were the entire plant (FL; 50%) prepared as decoction with oral administration, and leaves (FL; 37%) mostly cooked and eaten as soup. For the treatment of fever, informants mainly used the leafy stems (FL; 100%) which are frequently ground in water (FL; 85%), and administered both as an herbal bath (FL; 75%) and through the oral route (FL; 75%). For *S. leptocarpa*, similar to *H. indicum*, twelve uses were recorded, and the species was mostly used by herbal traders to treat hypertension (FL; 50%), and to facilitate childbirth (FL; 30%). For the treatment of hypertension, informants frequently use the entire plant (FL; 80%) as a decoction (FL; 100%) with oral administration (FL; 100%). To ease childbirth, herbal traders frequently exploit the leafy stems (FL; 66%) as a decoction (FL; 100%) through the oral route. *A. mexicana*; based on the full number of disease reported in the Table 4, the highest number of uses (15) was reported by herbal medicine traders for this species. The most common was the treatment of babies just after the fall of the umbilical cord (FL; 50%), and the entire plant was the most frequently used (FL; 77%) as a decoction (FL; 100%) through the oral route (FL; 100%). The treatment of jaundice was the second most recorded (FL; 33%), and informants mostly used the leafy stems (FL; 50%) as a decoction (FL; 100%) through oral administration (FL; 100%). The whole plant was also reported as used for that treatment (FL; 33%), as a decoction (FL; 100%) through oral administration (FL; 100%). The spiritual use to win conflicts was the third most recorded (FL; 22%), for which the respondents reported the chewing of fresh leaves of the species. *K. foetidissima*; eight different uses were recorded for *K. foetidissima* and the species was mostly reported in fighting against bad spirits (FL; 31%) by exploiting the leafy stems of the species (FL; 100%). Among the preparation, the leafy stems are sometimes either ground in water and administered orally, laid under a pillow or in a house corner. Furthermore, the spiritual use for good opportunities, the use for purification as well as the use of the species for self-protection were equally reported. For instance, to achieve good opportunities, leafy stems were mostly exploited (FL; 66%), pounded with soap or ground in water (FL; 50% for each) and used for bathing. This part was similarly reported (FL; 66%) for self-protection and was either pounded (FL; 50%) for oral administration or infused in coconut water (FL; 50%) for herbal bathing. For *P. pellucida*, eleven uses were reported, and the most common was its exploitation to implant a West African traditional religion called vodun, using the entire plant. Vodun implantation concerns its laying on the ground following some rituals. This was followed by the spiritual use for luck in which the entire plant was sometimes deposited under the traders' goods.

**Table 4.** Diversity of uses, parts, preparations, and routes of administration for *H. indicum*, *A. mexicana*, *S. leptocarpa*, *K. foetidissima*, *P. pellucida*.

| Social Conditions (FL%) * | Parts Used (FL%) | Preparation Modes (FL%) | Routes of Administration (FL%) |
|---|---|---|---|
| *H. indicum* | | | |
| Fever (29) | Leafy stems (100) | Ground in water (85) | Oral (75) |
| | | | Herbal bathing (75) |
| | | | Ointment (25) |
| | | Decoction (20) | Oral (100) |
| | | Ground with no water addition (40) | Oral (50) |
| | | | Ointment (50) |
| | Entire plant (20) | Decoction (100) | Oral (100) |
| Hypertension (47) | Leafy stems (25) | Decoction (100) | Oral (100) |
| | | Pounded (50) | Oral (100) |
| | Leaves (37) | Cooked as soup (100) | Oral (100) |
| | | Cooked as soup (100) | Oral (100) |
| | Entire plant (50) | Decoction (100) | Oral (100) |
| | | Infusion in hot water (25) | Oral (100) |
| Measles (5) | Leaves (100) | Ground with no water addition (100) | Oral (100) |
| Stomach infections (17) | Entire plant (100) | Decoction (100) | Oral (100) |
| | Leafy stems (33) | Decoction (100) | Oral (100) |
| Spiritual use for protection (11) | Seeds (50) | Burnt and powdered (100) | Oral (100) |
| | Entire plant (50) | Not reported | Not reported |
| Skin infections (5) | Entire plant (100) | Ground in water (100) | Herbal bathing (100) |
| Spiritual use for luck (5) | Flowers (100) | Chewed (100) | Oral (100) |

**Table 4.** *Cont.*

| Social Conditions (FL%) * | Parts Used (FL%) | Preparation Modes (FL%) | Routes of Administration (FL%) |
|---|---|---|---|
| Hypotension (11) | Dry leaves (50) | Decoction (100) | Oral (100) |
| | Fresh leaves (50) | Cooked as soup (100) | Oral (100) |
| | Entire plant (50) | Decoction (100) | Oral (100) |
| Vaginal infections (11) | Entire plant (100) | Decoction (100) | Personal hygiene (50) |
| | | | Oral (50) |
| Intestinal infections (5) | Entire plant (100) | Decoction (100) | Oral (100) |
| All infections (5) | Entire plant (100) | Decoction (100) | Oral (100) |
| Diabetes (5) | Not reported | Not reported | Not reported |
| *S. leptocarpa* | | | |
| To facilitate childbirth (30) | Leafy stems (66) | Decoction (100) | Oral (100) |
| | Entire plant (33) | Decoction (100) | Oral (100) |
| Infantile convulsion (10) | Entire plant (100) | Decoction (100) | Oral (100) |
| Ulcer (10) | Entire plant (100) | Decoction (100) | Oral (100) |
| Unresolved pregnancy (10) | Entire plant (100) | Decoction (100) | Oral (100) |
| Child failing to thrive (10) | Entire plant (100) | Decoction (100) | Herbal bathing (100) |
| Spiritual use to win conflicts (10) | Leaves (100) | Chewed (100) | Oral (100) |
| Jaundice (10) | Entire plant (100) | Decoction (100) | Oral (100) |
| Hypertension (50) | Leafy stems (20) | Decoction (100) | Oral (100) |
| | Entire plant (80) | Decoction (100) | Oral (100) |
| Lower fetal weight (10) | Entire plant (50) | Decoction (100) | Oral (100) |
| Spiritual use for a peaceful family (10) | Leafy stems (100) | Powdered (100) | Put in ember (100) |

**Table 4.** *Cont.*

| Social Conditions (FL%) * | Parts Used (FL%) | Preparation Modes (FL%) | Routes of Administration (FL%) |
|---|---|---|---|
| Hemorrhoid (10) | Entire plant (100) | Burnt and powdered (100) | Oral (100) |
| Spiritual use by Ifa priests (10) | Entire plant (100) | Not reported | Not reported |
| Lower fetal weight (10) | Entire plant (50) | Decoction (100) | Oral (100) |
| Spiritual use for a peaceful family (10) | Leafy stems (100) | Powdered (100) | Put in ember (100) |
| Hemorrhoid (10) | Entire plant (100) | Burnt and powdered (100) | Oral (100) |
| Spiritual use by Ifa priests (10) | Entire plant (100) | Not reported | Not reported |
| *A. mexicana* * | | | |
| Jaundice (33) | Dry seeds (16) | Powdered (100) | Oral (100) |
| | Dry Leaves (16) | Decoction (100) | Oral (100) |
| | Leafy stems (50) | Decoction (100) | Oral (100) |
| | Entire plant (33) | Decoction (100) | Oral (100) |
| | Fresh Leaves (16) | Infusion in hot water (100) | Oral (100) |
| Malaria (16) | Leafy stems (33) | Decoction (100) | Oral (100) |
| | Entire plant (66) | Decoction (100) | Oral (100) |
| Stomach aches (11) | Entire plant (100) | Decoction (100) | Oral (100) |
| Infantile stomach aches (16) | Leafy stems (100) | Decoction (100) | Oral (100) |
| Stomach infections of adults and kids (11) | Entire plant (100) | Decoction (100) | Oral (100) |
| Baby treatment after umbilical cord fall (50) | Leafy stems (22) | Decoction (100) | Oral (100) |
| | Entire plant (77) | Decoction (100) | Oral (100) |
| Mouth infections (5) | Entire plant (100) | Decoction (100) | Oral (100) |

**Table 4.** *Cont.*

| Social Conditions (FL%) * | Parts Used (FL%) | Preparation Modes (FL%) | Routes of Administration (FL%) |
|---|---|---|---|
| Spiritual use to win conflicts (22) | Leafy stems (25) | Not reported | Not reported |
| | Leaves (50) | Chewed (100) | Oral (100) |
| | Entire plant (25) | Not reported | Not reported |
| Infantile intestinal infections (5) | Leafy stems (100) | Decoction (100) | Oral (100) |
| | Entire plant (100) | Decoction (100) | Oral (100) |
| Intestinal infections of adults and children (5) | Entire plant (100) | Decoction (100) | Oral (100) |
| Spiritual use for luck (16) | Entire plant (100) | Planting at home (33) | Planting (100) |
| | | Pounded with soap mixture (66) | Herbal bathing (100) |
| | | Pounded (33) | Herbal bathing (100) |
| | | Infusion in spray (33) | Spraying (100) |
| Liver disorders (5) | Leafy stems (100) | Decoction (100) | Oral (100) |
| Hemorrhoid (5) | Entire plant (100) | Decoction (100) | Oral (100) |
| Intestinal cleaning of babies (5) | Leafy stems (100) | Decoction (100) | Oral (100) |
| All infections (5) | Leafy stems (100) | Decoction (100) | Oral (100) |
| Constipation (5) | Leafy stems (100) | Decoction (100) | Oral (100) |
| Liver disorders (5) | Leafy stems (100) | Decoction (100) | Oral (100) |
| Hemorrhoid (5) | Entire plant (100) | Decoction (100) | Oral (100) |
| Intestinal cleaning of babies (5) | Leafy stems (100) | Decoction (100) | Oral (100) |
| All infections (5) | Leafy stems (100) | Decoction (100) | Oral (100) |
| Constipation (5) | Leafy stems (100) | Decoction (100) | Oral (100) |
| *K. foetidissima* | | | |
| Spiritual use for good opportunities (18) | Leafy stems (66) | Pounded with soap mixture (50) | Herbal bathing (100) |
| | | Ground in water (50) | Herbal bathing (100) |

**Table 4.** *Cont.*

| Social Conditions (FL%) * | Parts Used (FL%) | Preparation Modes (FL%) | Routes of Administration (FL%) |
|---|---|---|---|
| Purification (18) | Leafy stems (66) | Infusion in sea water (100) | Sprinkled flowerbed (100) |
| | Entire plant (66) | Infusion in sea water (50) | Sprinkled flowerbed (100) |
| | | | Herbal bathing (100) |
| | | Pounded (50) | Scattered on the ground (100) |
| Fighting against bad spirits (31) | Leafy stems (100) | Ground in water (20) | Oral (100) |
| | | Laid under the pillow (20) | Laid under the pillow (100) |
| | | Infusion in palm kernel oil (20) | Ointment (100) |
| | | Laid in a house corner (20) | Laid in a house corner (100) |
| Fever (6) | Leafy stems (100) | Ground in water (100) | Herbal bathing (100) |
| | | Ground in water with addition of palm oil (100) | Oral (100) |
| Convulsion (12) | Leafy stems (100) | Pounded with juice extraction (100) | Oral (100) |
| Spiritual use to implant a vodun (6) | Leafy stems (100) | Not reported | Not reported |
| Spiritual use for self-protection (18) | Leafy stems (66) | Pounded with extraction of juice (50) | Oral (100) |
| | | Infusion in coconut water (50) | Herbal bathing (100) |
| | Entire plant (33) | Not reported | Not reported |
| Measles (12) | Fresh leaves (100) | Pounded with extraction of juice (50) | Ointment (100) |
| | | Infusion in ethanol (50) | Oral (100) |
| *P. pellucida* | | | |
| Cleaning of bodily impurities (7) | Entire plant (100) | Infusion in water (100) | Herbal bathing (100) |
| Spiritual use for a peaceful family (7) | Entire plant (100) | Infusion in water (100) | Oral (100) |

| Social Conditions (FL%) * | Parts Used (FL%) | Preparation Modes (FL%) | Routes of Administration (FL%) |
|---|---|---|---|
| Headaches (7) | Entire plant (100) | Powdered (100) | Head scarification (100) |
| Spiritual use to implant a vodun (23) | Entire plant (100) | Not reported | Not reported |
| Spiritual use for luck (15) | Entire plant (100) | Deposited under traders' goods (50) | Deposited under traders goods (100) |
| Women's hot flush (7) | Entire plant (100) | Decoction (100) | Oral (100) |
| Stiffness (7) | Entire plant (100) | Decoction (100) | Oral (100) |
| Spiritual use for purification (7) | Leafy stems (100) | Ground in water (100) | Herbal bathing (100) |
| Azoospermia (7) | Entire plant (100) | Pounded (100) | Oral (100) |
| Eye burns (7) | Leaves (100) | Ground to extract the juice (100) | Ocular (100) |
| Infantile fever (7) | Entire plant (100) | Ground in water (100) | Herbal bathing (100) |

* Disease should not be double counted and full list of 15 diseases considered from Table 4. for *A. mexicana* are Jaundice, malaria, stomach aches, stomach infections, treatment for babies after umbilical cord fall, mouth infections, spiritual use to win conflicts, spiritual use for luck, intestinal infections for kids, liver disorders, hemorrhoid, intestinal cleaning, all infections, constipation and intestinal infections for kids and adults.

Similar to the present findings, the hypotensive activity of *H. indicum* has been reported in many other studies, [45–47] including in vivo tests. Other properties of *H. indicum* include its antituberculosis activity using the essential oil from the aerial parts of the species [18]. The frequent use of the entire plant of this species threatens its survival. The use of its leaves as soup to treat for to hypotension confirms the diversity of knowledge held by herbal medicine traders. Quenum et al. [31] reported a reduction in blood pressure in Wistar rats treated with a combination of *S. leptocarpa*, *Garcinia kola* Heckel and *Ocimum americanum* L. This matches with the treatment of hypertension using *S. leptocarpa* recorded in this study. However, further pharmacological studies on the hypotensive effects and toxicity of extracts of *S. leptocarpa* are required. Attah et al. [48] studied the uterine contractility of plants used in Nigerian ethnomedicine to facilitate childbirth. Similar research with *S. leptocarpa* in Beninese ethnomedicine is recommended since pregnancy is a vulnerable condition. *K. foetidissima*, mainly recorded for spiritual use in this study, is less exploited in other parts of the world than other species of Cucurbitaceae [49]. However, the second rank species in terms of demand in this study prove that it plays a key role in traditional use in Benin. The spiritual use of plants is very common in the country [3,4] and confirms the connection of African communities with traditional beliefs since time immemorial. *P. pellucida* was also recorded mainly for its spiritual use in this research. Elsewhere, the species is also used against fever, headaches, and conjunctivitis [24]. Regarding the pharmaceutical properties, the analgesic, anti-inflammatory antihyperglycemia and hypotensive activities of *P. pellucida* species were reported [26]. It is also used against fever, headaches, and conjunctivitis [24]. The exploitation of the entire plant recorded in this research threatens species survival.

*A. mexicana* was recorded mainly in the treatment of babies after the fall of the umbilical cord, corroborating the findings of Brahmachari et al. [50] who reported the utilization of the species to treat microbial infections and inflammations. Similar to the present findings,

the exploitation of *A. mexicana* against jaundice has been widely stated [50,51]. In terms of pharmaceutical relevance, Chang et al. [13] stated that the methanolic extract of air-dried whole plants of *A. mexicana* exhibited potent anti-HIV activity. The rank of *A. mexicana* as the most demanded of the species studied, together with the highest number of use reports for this species, confirm its important role in traditional medicine in Benin. The use of the entire plant of *A. mexicana* is a threat to the species' survival. As a result, further investigations should be focused on the conservation tools and strategies for the species throughout Benin. Furthermore, there is an urgent need to promote planting of the species throughout the country.

Notably, species Value in Use (UV), similarity, and the contribution of traders to species conservation were also examined. Amongst the five species, *S. leptocarpa* exhibited the highest Use Value (UV: 0.6; Total citations; 20 with 12 use reports). It was followed by *P. pellucida* (UV: 0.42; 26 citations and 11 use reports), *A. mexicana* (UV: 0.41; 36 citations and 15 use reports). *H. indicum* (UV: 0.35; 34 citations and 12 use reports), and *K. foetidissima* had the lowest UV (UV: 0.25; 32 citations and 8 use reports). With regard to the similarity in terms of use reports, no pair of species exhibited a high similarity, since all RSI values were lower than 50% (Table 5). However, noticeable values of RSI were obtained between *H. indicum*, *K. Foetidissima* (33%), and *A. mexicana* (27%). Although lower than 50%, a value of 26% was obtained for RSI between *K. foetidissima* and *P. pellucida*.

**Table 5.** Shared social conditions of the study species and the Rahman Similarity Index (RSI).

| Plant Samples | *H. indicum* (*n* = 12) | *S. leptocarpa* (*n* = 12) | *A. mexicana* (*n* = 15) | *K. foetidissima* (*n* = 8) | *P. pellucida* (*n* = 11) |
|---|---|---|---|---|---|
| *H. indicum* (*n* = 12) | | | | | |
| *S. leptocarpa* (*n* = 12) | -Hypertension RSI = 4.34% | | | | |
| *A. mexicana* (*n* = 15) | -Stomach infections -Spiritual use for luck -Intestinal infections -All infection RSI = 27.39% | -Spiritual use to win conflicts -Jaundice -Hemorrhoid RSI = 12.5% | | | |
| *K. foetidissima* (*n* = 8) | -Spiritual use -Fever -Spiritual use for self-protection -Spiritual use for luck -Measles RSI = 33.33% | RSI = 0% | -Spiritual use for good opportunities = spiritual use for luck RSI = 4.54% | | |
| *P. pellucida* (*n* = 11) | -Fever = infantile fever -Spiritual use for luck RSI = 9.52% | RSI = 0% | -Spiritual use for luck RSI = 4% | -Spiritual use to implant a vodun -Spiritual use for good opportunities = spiritual use for luck -Spiritual use for purification -Fever RSI = 26.66% | |

Overall, the majority of traders did not plant the five research species. The data regarding planting were, for *H. indicum*, planting RFC, 11%; no planting RFC, 66%; naturally grown at home and conserved RFC, 17%. For *S. leptocarpa* they were naturally grown at home and conserved RFC, 11%; no planting RFC, 44%. For *A. mexicana*, planting RFC, 11%; no planting RFC, 72 %; naturally grown at home and conserved RFC, 17%. For *K. foetidissima*, planting RFC, 0%; naturally grown in own gardens and conserved RFC, 5%; no planting, 100%. For *P. pellucida*, planting RFC, 22%; no planting RFC, 44%; naturally grown at home and conserved RFC, 6%. It is important to highlight that all traders who reported the collection of *K. foetidissima* from their own gardens (Table 1) confirmed its natural occurrence. Moreover, all traders who confirmed the planting of the research species did it for trade and personal use.

Although ranked as the least in demand by customers, *S. leptocarpa* exhibited the highest UV. *P. pellucida*, ranked as fourth in terms of demand, was the second in relative importance. However, *A. mexicana* ranked as the most demanded and with the highest number of use reports was only third in terms of relative importance. These findings show the importance of UV in comparative ethnobotanical studies. In fact, a taxon may not exhibit the highest use reports but still be relatively the most important in terms of uses [43]. The low similarity between species in terms of use reports means that the species studied cannot serve as alternative herbal remedies for each other for considerable numbers of recorded social conditions. Overall, the research species were not planted by herbal medicine traders. In fact, the species are very useful in herbal remedies in the country, and traders benefit from income generated by their commerce without contributing to their conservation and sustainable exploitation. For instance the exploitation of the entire plants of *P. pellucida*, *A. mexicana* and *H. indicum* recorded in this research are potential threats to the species conservation. However *H. indicum* is a common weed in Benin so the protection should not be the aim but possibly control by its utilization for medicinal purposes. Although *A. mexicana* is becoming rare in Southern Benin, it is elsewhere a widespread annual weed primarily associated with agricultural crops and wastelands. It is a major weed of a number of crops in the tropics and warm temperate regions and is persistent as it produces a seed bank. In India in particular, the species is a health hazard and because of its prickliness, is a nuisance to subsistence farmers. In South Africa the seeds of *A. mexicana* have been declared as 'noxious' as its seeds or bits of seeds may represent a hazard to human or animal health when consumed [52]. It is reported as invasive in many countries in Asia, Africa and other parts (https://www.cabi.org/isc/datasheet/6878 accessed on 29 June 2022); *H. indicum* occurs as a weed in Africa and other parts of the world (https://www.cabi.org/isc/datasheet/26899 accessed on 29 June 2022).

Mahunnah [53] highlighted that medicinal plants are important worldwide and that the conservation problem is a major threat to the rich biodiversity of medicinal plants in Africa. This point of view was later confirmed by the findings of Nigro et al. [54], who stated that the material from which useful compounds are extracted is not always cultivated or wild-harvested in a sustainable way. Moreover, plants known as medicines years ago no longer exist today [6]. Contrary to this evidence of threats, indigenous people's interest in the use of medicinal plants has always been based on the wrong assumption that these plants will be available on a continuing basis [55]. All of these observations and findings reinforce the need to sensitize indigenous people, to promote the conservation and planting of the research species, as well as undertaking many other investigations on herbaceous medicinal plants in Benin.

## 4. Conclusions

The five species studied are alien plants used for many traditional purposes. Since *S. leptocarpa* is an invasive species and *H. indicum* is a common weed in Southern Benin, and taking into account the availability reported by the traders on *S. leptocarpa*. we suggest the management of these two species through the trading and other medico-magical utilization. Regarding the scarcity reported with regard to *A. mexicana*, the use of the entire plant in

addition to the fact that it is becoming rare in addition to *P. pellucida* in Southern Benin, their conservation is suggested by planting in commercial and home gardens. Since *K. foetidissima* is a spontaneous rare weed, its preservation and sustainable utilization are recommended in this paper.

**Author Contributions:** Conceptualization, H.O.D.-Y. and F.G.V.; methodology, H.O.D.-Y. and F.G.V.; software, H.O.D.-Y., F.G.V. and A.K.; validation, H.O.D.-Y., F.G.V., A.K. and B.S.; formal analysis, H.O.D.-Y., F.G.V. and A.K.; investigation, H.O.D.-Y. and F.G.V.; resources, H.O.D.-Y. and F.G.V.; data curation, H.O.D.-Y. and F.G.V.; writing—original draft preparation, H.O.D.-Y. and F.G.V.; writing—review and editing, H.O.D.-Y., F.G.V., A.K. and B.S.; visualization, H.O.D.-Y., F.G.V. and A.K.; supervision, B.S.; project administration, B.S.; funding acquisition, all authors afforded the study cost. All authors have read and agreed to the published version of the manuscript.

**Funding:** This research received no external funding.

**Institutional Review Board Statement:** The study was conducted in accordance with the Declaration of Helsinki and approved by research institues. A research proposal was submitted and approved by the Head of our research institute, Laboratory of Applied Ecology of Faculty of Agronomic Sciences, University of Abomey-Calavi, Benin. He participated in this study; he approved the ethics for ethnobotanical investigations through market surveys. We received the ethical clearance of the Scientific Council of the University of Abomey-Calavi (N°145-2021/UAC/VR-RU/SCS/SA on 14 April 2021) for the same research proposal which already served for a research undertaken on other herbaceous species.

**Informed Consent Statement:** The aim of our research was explained to the responsible of each surveyed market. They provided the verbal content to conduct the present investigation. Similarly, the aim was explained to each trader to get their verbal consent to participate in the research and to approve the publication of this paper. Consent was obtained from all subjects involved in this study.

**Data Availability Statement:** The ethical clearance and the plant certification are available at Diversity (MDPI). All data supporting various calculations and results reported in the text are available within the 5 detailed tables which are also available at Diversity (MDPI). Furthermore, any scientist interested in conducting similar study from any part of the world can communicate with authors for free technical assistance.

**Acknowledgments:** Authors are deeply grateful to Phil Harris from Coventry University (England) who read and edited the first draft of this paper and provided relevant comments for its improvement. Thanks are also due all reviewers and editors who spent their valuable time on this paper. All herbal traders who participated in this research are thanked.

**Conflicts of Interest:** The authors declare no conflict of interest.

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
