# Peer review of "Application of Ethnobotanical Indices in the Utilization of Five Medicinal Herbaceous Plant Species in Benin, West Africa"

_diversity, doi:10.3390/d14080612_

Round 1
Reviewer 1 Report
Some points should be added:
1- Update the introduction section with more recent articles on the previous studies in the same area of your study.
2- You should add a location map to the section of the study area and more information about climate and land use in the area supported also with photos for the most common markets which have been used in the survey to be more realistic study.
3- The authors must add more information about the vegetation of the area shown the source of the medicinal plants which already used in the markets.
4- The Photos of studied species i.e. used species in the market, should be increased with a clear and bright form.
5-The tables must be reduce in size and you should create and informatic graph which can help reader and summarize the data better in the paper.
Author Response
Response to the Reviewer 1:
C1: Update the introduction section with more recent articles on the previous studies in the same area of your study.
R1: The introduction has been updated with more recent articles on previous work in the same field as our work.
C2: You should add a location map to the section of the study area and more information about climate and land use in the area supported also with photos for the most common markets which have been used in the survey to be more realistic study.
R2: For a more realistic study, more information about climate in the area has been added to the study area section, supplemented by a location map showing the vegetation and soil distribution. Since there are already a lot of tables and two figures in the manuscript, we did not add photographs of the surveyed markets.
C3: The authors must add more information about the vegetation of the area shown the source of the medicinal plants which already used in the markets.
R3: More information about the vegetation of the area added to the article.
C4: The Photos of studied species i.e. used species in the market, should be increased with a clear and bright form.
R4: Photos have been added as you want and made more understandable.
C5: The tables must be reduce in size and you should create and informatic graph which can help reader and summarize the data better in the paper.
R5: Since there is a lot of data in the tables, this is the most descriptive and organized form that can be given. Adequate explanation has already been given in the article as a support.
*Dear reviewer 1, we made all the corrections you mentioned in the MS. Thank you for your valuable comments.
Reviewer 2 Report
The authors reported an original study on ethnobotany of species grown in Africa. The ethnographic and anthropological aspects are very important when studying new therapeutic applications for plant species used for religious and animistic purposes.
However, the authors gave a general overview of plant species neglecting the botanical and agronomic aspects. Ecological conditions of growth, harvesting and conservation of plants are important parameters. It is also not clear which parts of the plants are used. Geographic distribution of use. All these aspects, but not only them, should be better developed. In this regard, see: Bonini SA et al. Cannabis sativa: A comprehensive ethnopharmacological review of a medicinal plant with a long history. J Ethnopharmacol. 2018.
I suggest eliminating the phrases that refer to "fight evil spirits", it would be better to use "religious and / or animist purposes".
Author Response
Response to the Reviewer 2:
C1: The authors reported an original study on ethnobotany of species grown in Africa. The ethnographic and anthropological aspects are very important when studying new therapeutic applications for plant species used for religious and animistic purposes. However, the authors gave a general overview of plant species neglecting the botanical and agronomic aspects. Ecological conditions of growth, harvesting and conservation of plants are important parameters. It is also not clear which parts of the plants are used. Geographic distribution of use. All these aspects, but not only them, should be better developed. In this regard, see: Bonini SA et al. Cannabis sativa: A comprehensive ethnopharmacological review of a medicinal plant with a long history. J Ethnopharmacol. 2018.
R1: Information related to the origins and distribution of our research species were provided. This information help readers know where the species may be found and used used. Brief botanical description, ecological preferences of the species were also provided. It was stated which parts of the plants were used. Information was given about the geographical distribution of use by quoting other studies through the discussion of the results.
C2: I suggest eliminating the phrases that refer to "fight evil spirits", it would be better to use "religious and / or animist purposes".
R2: Removed "fight evil spirits" phrases and corrected them to "religious and/or animist purposes".
*Dear reviewer 2, we made all the corrections you mentioned in the MS. Thank you for your valuable comments.
Reviewer 3 Report
Comments:
The manuscript titled “Application of Ethnobotanical Indices in the Utilization of Five Medicinal Herbaceous Plant Species in Benin, West Africa” should be improved, in particular, English editing/proofreading is strongly recommended to enhance the overall manuscript presentation. Numerous errors were found throughout the manuscript.
Many technical errors were detected; some are mentioned but not all. Authors are recommended to check the whole manuscript for consistency and accuracy.
Line 73: he?
Line 76: EC50 value. 1.77 μg/mL?
Line 77: Reconstruct the whole sentence
Line 95-99: Please explain the diffusion technique is related to the activity. How could it be effective without showing any inhibition values?
Line 103-107: Reconstruct the sentence
Line 146-148: 1.010 or 1,010? 8.595 or 8,595?
Line 163: in r. ?
Line 179-180: Reconstruct the sentence
Line 181-182: Reconstruct the sentence
Line 205-206: Symbols are missing to indicate the meaning of the equation
Line 210-211: Symbols are missing to indicate the meaning of the equation
Line 220-221: Symbols are missing to indicate the meaning of the equation
Line 237-238: Reconstruct the sentence
Line 241: 50% of what?
Line 300: ground in water (FL; 80%), but in Table, it showed 85%!
Line 306: There are more than 15 uses in Table 4 for A. mexicana
Line 312: Please be consistent in using “entire plant” and “whole plant” throughout the manuscript.
Line 333: Plant species names should be in italic.
Line 370: Species Value in Use (UV) is first mentioned with its full name. However, its abbreviation has been used without introducing the full name in Abstract and Introduction.
Introduction is lengthy and not organised, many plant species are mentioned here and there. At one point, it would be confusing the plant extract is referring to which species. Some statements Introduction were found repeating in Results and Discussion.
Abbreviations are used without giving the full names when they are first mentioned in text. And, once abbreviations were used, please consistently using them throughout the text. Do not repeat the full name again.
In Methodology, Kpassê was mentioned in study area; but Godomey (not mentioned in study area) is mentioned in Line 186.
Table 1 & 2: It should be H. indicum and etc.
Table 4 is not organised, it is hard to refer. Section/horizontal lines are recommended to divide each social condition.
In conclusion, authors are recommended to conclude the significance of the present study.
Reference style is not consistent. Please revise.
Author Response
Response to the Reviewer 3:
C1. Line 73: he?
R1. HeLA indicated in line 73 is involved in the expression of the cancer cell and HepG is human liver cancer cell line.
C2. Line 76: EC50 value. 1.77 μg/mL?
R2. Expression was corrected. A better passage has been suggested in the text.
C3. Line 77: Reconstruct the whole sentence
R3.We reconstruct the sentence and we wrote the following sentence: All parts of the H. indicum plant are claimed to have medicinal properties. The entire plant contains alkaloids such as heliotrin, laciocarpin, indisine, acetyl indices, indisine, indisine-N-oxide, retronesin, trachelantamide.
C4. Line 95-99: Please explain the diffusion technique is related to the activity. How could it be effective without showing any inhibition values?
R4. Although the disc diffusion method is a semi-numerical method, it is advantageous in terms of not serial dilution and easy interpretation, and the antibacterial effect was studied using this technique in the study.However, we add inhibiton zone values immediately: They found that the technique was effective against the bacteria tested, and the maximum and minimum inhibition zones were Pseudomonas aeruginosa (10mm) and Vibrio cholera (3mm) on the leaf, respectively.
C5. Line 103-107: Reconstruct the sentence
R5. We reconstruct the sentence and we wrote the following sentence: P. pellucida extracts, fractions or isolated components have been shown to have analgesic, anti-inflammatory, antipyretic, antioxidant, antihyperglycemia, antihyperuricemia, burn healing, depressant effect, gastric protective, hypotensive, cytotoxic, antimicrobial, antidisease cell, lipase inhibitor. However, it has fibrinolytic and thrombolytic, antidiarrheal, antiosteoporotic, antihyperglycemia activities.
C6. Line 146-148: 1.010 or 1,010? 8.595 or 8,595?
R6. Corrected to 1,010 and 8,595 in the article.
C7. Line 163: in r. ?
R7. The expression is edited correctly with, local language Fon.
C8. Line 179-180: Reconstruct the sentence
R8. We reconstruct the sentence and we wrote the following sentence: The aim of the research was explained to the officials responsible of each market from whom the approval to conduct the surveys was sought. So the verbal informed consent was obtained from all the market responsibles.
C9. Line 181-182: Reconstruct the sentence
R9. We reconstruct the sentence and we wrote the following sentence: Similarly, the aim of the study was explained to each trader from whom we obtained the verbal consent to participate in the research.
C10. Line 205-206: Symbols are missing to indicate the meaning of the equation
R10. Symbols were added and shows as “where ni is the number of informants who mentioned the concerned species, thing or aspect and N is the total number of traders surveyed” , and formula is,
, (1)
C11. Line 210-211: Symbols are missing to indicate the meaning of the equation
R11. Symbols were added and shows as “where Ip is the number of informants who mentioned the use of a species for a specific disease, disorder or magic issue and Iu is the total number of informants who mentioned the species for any use” and formula is,
, (2)
C12. Line 220-221: Symbols are missing to indicate the meaning of the equation
R12.Symbols were added and shows as “where nd is the total number of use reports (diseases, disorders or magic utilization) mentioned for a species and Iu is the total number of informants who mentioned the species for any use and formula is,
, (3)
Symbols were added and shows as “This formula is similar to the Jaccard Similarity Index. Rahman et al. (2019) considered an ailment recorded in two communities and treated with a number of medicinal plant species, species commonly used in both communities. In the present research, this approach was adapted to similarity of uses of pairs of species with, respectively, na and nb number of use reports in species a and b, respectively, and the number of common uses recorded for the two species as nc. and formula is,
, (4)
C13. Line 237-238: Reconstruct the sentence
R13. We reconstruct the sentence and we wrote the following sentence: All respondents confirmed the trade of A. mexicana while the majority of them, respectively 94 %, 88 %, and 72 %, traded H. indicum, K. foetidissima and P. pellucida.
C14. Line 241: 50% of what?
R14. The expression has been edited as 50% of informants.
C15. Line 300: ground in water (FL; 80%), but in Table, it showed 85%!
R15. Edited by changing it to 85%.
C16. Line 306: There are more than 15 uses in Table 4 for A. mexicana
R16. Diesase should not be double counted and full list of diesease were provide the concerned table 4. List of diseases for A. mexicana are: Jaundice, malaria, stomach aches, stomach infections, baby treatment after umblical cord fall, mouth infections, spiritual use to win conflicts, spiritual use for luck, intestinal infections for kids, liver diorders, hemorrhoid, intestinal cleaning, all infections, constipation and intestinal infections for kids and adults.
C17. Line 312: Please be consistent in using “entire plant” and “whole plant” throughout the manuscript
R17. Whole was change with entire.
C18. Line 333: Plant species names should be in italic.
R18. Species name italicized
C19. Line 370: Species Value in Use (UV) is first mentioned with its full name. However, its abbreviation has been used without introducing the full name in Abstract and Introduction.
R19. Value in Use is given with an explanation on first use in abstract.
C20. Introduction is lengthy and not organised, many plant species are mentioned here and there. At one point, it would be confusing the plant extract is referring to which species. Some statements Introduction were found repeating in Results and Discussion.
R20. The introduction part of the article has been shortened by organizing. The confusion created by the plant extracts has been cleared. The studies in the introduction were arranged so that the repetition that could arise in the conclusion and discussion was avoided.
C21. Abbreviations are used without giving the full names when they are first mentioned in text. And, once abbreviations were used, please consistently using them throughout the text. Do not repeat the full name again.
R21. In the first use of abbreviations in the whole text, the full name is written as you want, then the abbreviation is used.
C22. In Methodology, Kpassê was mentioned in study area; but Godomey (not mentioned in study area) is mentioned in Line 186.
R22. The sentence arrnaged. Godomey market was mentioned.
C23. Table 1 & 2: It should be H. indicum and etc.
R23. Tables were corrected to abbreviate the species name.
C24.Table 4 is not organised, it is hard to refer. Section/horizontal lines are recommended to divide each social condition.
R24. Table 4 has been modified as you want us to do.
C25. In conclusion, authors are recommended to conclude the significance of the present study.
R25. We added sentences to highlight the impotance of the study species and the significance of our study.
C26. Reference style is not consistent. Please revise.
R26. References revised.
*Dear reviwer 3, we made all the corrections you mentioned in the MS. Thank you for your valuable comments.
Response to the editorial comments:
C1. Several of the plant species are not native to the Africa.This should be well-differentiated and emphasized in the text because it correlates to traditional utilization.It is not possible to expect a well-established local traditional application for a plant that is not native
R1. We have added the native zones of each research species and emphasized that all of them are not native to Africa. Contrary to the reviwer’s comment stating that a non- native species can’t have ethnobotanical uses, we mentioned that all the species are widely naturalized in Benin where people collect them for ethnobotanical and medico-magic uses since decades.
*Dear reviewer, my co-authors and I made all the corrections you mentioned in the MS. Thank you for your valuable comments.
Round 2
Reviewer 1 Report
the only point I can say if you can draw a better map for the study area with a clear line and colors it would be better and showing the north direction as well.
Author Response
Dear reviewer,
Thank you